# Consumption of Omega-3 and Maintenance and Incidence of Depressive Episodes: The ELSA-Brasil Study

**DOI:** 10.3390/nu14153227

**Published:** 2022-08-07

**Authors:** Renata da Conceição Silva Chaves, Odaleia Barbosa Aguiar, Arlinda B. Moreno, André R. Brunoni, Maria del Carmem B. Molina, Maria Carmen Viana, Isabela Bensoñor, Rosane H. Griep, Maria de Jesus Mendes da Fonseca

**Affiliations:** 1Department of Food and Nutrition, Piquet Carneiro Polyclinic, Universidade do Estado do Rio de Janeiro, Rio de Janeiro 20950-000, Brazil; 2Department of Applied Nutrition, Universidade do Estado do Rio de Janeiro, Rio de Janeiro 20950-000, Brazil; 3Department of Epidemiology and Quantitative Methods in Health, Escola Nacional de Saúde Pública, Fundação Oswaldo Cruz, Rio de Janeiro 21040-900, Brazil; 4Laboratory of Neurosciences, Department and Institute of Psychiatry, Universidade de São Paulo, São Paulo 05508-220, Brazil; 5Graduate Studies Program in Collective Health, Universidade Federal do Espírito Santo, Vitória 29075-910, Brazil; 6Department of Social Medicine, Universidade Federal do Espírito Santo, Vitória 29075-910, Brazil; 7Department of Internal Medicine, University of São Paulo (USP), São Paulo 05508-070, Brazil; 8Laboratory of Education in Environment and Health, Instituto Oswaldo Cruz, Fundação Oswaldo Cruz, Rio de Janeiro 21040-900, Brazil

**Keywords:** dietary intake, incidence, depression, omega-3, fatty acids

## Abstract

Depression affects 264 million persons in the world, accounting for some 4.3% of the global burden of disease. Current studies indicate that the decrease in the consumption of omega-3 food sources is associated with the increasing incidence of depression. The study aims to assess the association between the consumption of omega-3 and the maintenance and incidence of depressive episodes in adults (39–64 years) and elderly adults (>65 years). This was a longitudinal study using data from the baseline and first follow-up wave of the Longitudinal Study of Adult Health (ELSA-Brasil). Depressive episodes were obtained with the *Clinical Interview Schedule Revised* (CIS-R), and food consumption was measured with the Food Frequency Questionnaire (FFQ). Logistic regression was used to analyze associations between the consumption of omega-3 and depressive episodes. Fatty acids from the omega-3 family showed a protective effect against the maintenance of depressive episodes. In relation to incidence, the estimates suggest that the higher the consumption of omega-3 acids, the lower the risk of developing depressive episodes, and significant associations were found between the consumption of omega-3 and alpha-linolenic acid. Dietary consumption of omega-3, DHA, EPA, DPA, and alpha linolenic fatty acids may have a protective effect against the maintenance and incidence of depressive episodes.

## 1. Introduction

The *Global Burden of Disease Study* (GBD) (2018) estimated that, in 2017, depression affected more than 264 million persons worldwide, ranking as the world’s third leading cause of years lived with disability in all age groups, with a prevalence greater than 10% in all the regions studied [1,2]. According to the World Health Organization (WHO), in 2016, depression accounted for approximately 4.3% of the global burden of disease and was directly related to quality of life, health, and work [3]. Due to the impact on the global burden of disease, the World Health Organization proposed the Comprehensive Mental Health Action Plan (2013–2020), extending measures that contribute to the prevention and treatment of psychiatric disorders to 2030 [4].

There are currently various types of treatments for the prevention and mitigation of depression, featuring psychological, psychosocial, psychiatric, pharmacological, and dietary forms that can be combined or used alone [5,6,7,8]. Diet acts in the treatment of noncommunicable chronic diseases and has been identified as a protective factor against mental disorders [5,9,10]. A healthier diet that contains specific nutrients, especially those related to the modulation of the neurotransmitters serotonin and dopamine, such as long-chain fatty acids from the *n*-3 (omega-3) family, can contribute positively to the prevention and treatment of depression [6,7,11]. Although the impact of omega-3 intake on the pathophysiology and symptoms of depression is not totally established, cross-sectional studies and metanalyses indicate that lower consumption of omega-3 food sources is associated with increased incidence of depression [12,13,14,15,16].

Omega-3 fatty acids have several functions in the body and metabolism, exhibiting a disease protection factor due to their anti-inflammatory, antithrombotic, and anti-arteriosclerotic effects. Additionally, Omega-3 fatty acids act on the modulation of serotonin and allow for an increase of the availability of this neurotransmitter in the synaptic cleft, which is essential for neural functioning [17,18,19]. Changes in the action, availability and serum production of serotonin and dopamine are related to the pathophysiology of depression, although all the mechanisms involved are not elucidated [14,20].

Some of the mechanisms listed in the literature explain the relationship between omega-3 and depression. Among them, it is highlighted that, despite the reduction in the amount of these neurotransmitters mentioned above due to the reduction of neuromodulation arising from the low serum level of omega-3 fatty acids, the actions of the reuptake pump and the enzymes involved in the degradation of these substances remain unchanged. Consequently, in depression, there is a lower uptake of neurotransmitters by the receptor neuron, effecting the functioning of the Central Nervous System (CNS) due to the inadequate level of neurotransmitters uptaken [6].

In addition, this imbalance may come from a deregulation of neurotransmission due to the instability of the neural cell membrane caused by an omega-3 deficit [22]. Generally, this destabilization occurs in situations of stress or inflammation, where some enzymes are attached to the cell membrane, removing fatty acids from phospholipids, thereby deregulating serotonin uptake receptors and other neurotransmitters [21,22].

There is no consensus on the subject in the scientific literature, with some studies emphasizing that the relationship between nutrient intake and mental disorders is not entirely elucidated, showing inconsistences in this relationship [11,23], while others point to dietary interventions as a potential key factor in the treatment of depression in adults [23,24,25,26].

The number of studies on the association between depression and dietary intake has increased rapidly, specifically regarding omega-3 fatty acids, due to this nutrient’s potential effects on the cell membrane’s maintenance and fluidity, the formation of new neurons, neuromodulation, and neurotransmission, especially in the context of rising prevalence rates and incidence of depression and the latter’s impact on the global burden of disease [3,10,15,27,28].

Despite the topic’s relevance, there are few longitudinal studies that address this association. Thus, the current study’s objective was to assess the association between consumption of omega-3 and the maintenance and incidence of depressive episodes in young and elderly adults participating in the Brazilian Longitudinal Study of Adult Health (ELSA-Brasil).

## 2. Materials and Methods

### 2.1. Study Design, Data Collection, and Study Population

Data were obtained from the Brazilian Longitudinal Study of Adult Health (ELSA- Brasil). This is a multicenter prospective cohort study consisting of public employees of both sexes ranging from 35 to 74 years of age (at baseline), belonging to six Brazilian teaching and research institutions (Oswaldo Cruz Foundation—Rio de Janeiro, University of São Paulo, Federal University of Bahia, Federal University of Minas Gerais, Federal University of Espírito Santo, and Federal University of Rio Grande do Sul) [29].

The baseline for ELSA-Brasil occurred from 2008 to 2010 and enrolled 15,105 individuals, and the first follow-up wave occurred from 2012 to 2014 and assessed 14,014 participants from the baseline. The two data collections were performed by trained and certified interviewers and verifiers according to criteria stipulated by a Quality Control Committee [30]. Participants answered multidimensional questionnaires consisting of socioeconomic questions, clinical history, occupational history, access to the health system, psychosocial and nutritional factors, smoking, consumption of alcohol and medications, physical activity, cognitive function, and mental health [31].

The current study included eligible participants of both sexes from the baseline and first follow-up wave, totaling 13,879 individuals. We excluded individuals that did not participate in the first follow-up wave of ELSA-Brasil (*n* = 1091), those who reported consumption of omega-3 supplements (*n* = 43), and subjects with incomplete or missing data related to mental and dietary health questions (*n* = 92) (Figure 1).

The article’s outcome was analyzed considering two conditions: maintenance of depressive episodes, that is, participants who presented depressive episodes at baseline and participated in the first follow-up wave, and who were classified again as either presenting depressive episodes (maintenance) or lack of maintenance. The total number of participants presenting depressive episodes at baseline was 582, 158 of whom presented depressive episodes at the first follow-up wave and 424 of whom did not. Incidence of depressive episodes was calculated by excluding participants who presented depressive episodes at baseline and considering those who presented depressive episodes at the first follow-up wave, namely 496 participants (Figure 1).

### 2.2. Study Variables

#### 2.2.1. Outcomes

The *Clinical Interview Schedule Revised* (CIS-R) was used to assess the outcomes: maintenance of depressive episodes and incidence of depression. The instrument was developed by Lewis et al. and allows for the assessment of occurrence, severity, and duration of symptoms of common mental disorders in the previous seven days, using a structured interview [32].

The instrument was translated and adapted to Brazilian Portuguese and consists of 14 sections corresponding to symptoms that caused suffering and alterations in routine activities in the previous week: anxiety, phobia, panic, compulsions, obsessions, physical symptoms, fatigue, depression, depressive ideas, irritability, lack of concentration and forgetfulness, altered sleep, preoccupation with body functioning, and general preoccupations [33].

The CIS-R was applied by trained and certified interviewers, in a face-to-face interview as part of the whole ELSA-Brasil questionnaire. Depressive episodes were computed by an algorithm developed by Lewis et al. according to ICD-10 criteria (F32.xx) for depressive episodes (mild, moderate with symptoms, moderate without symptoms, severe with symptoms, severe without symptoms) [33]. These depressive episodes classified by the CIS-R were grouped in a dichotomous variable as presence or absence of depressive episodes. There were no clinical diagnoses for depression assessed by clinicians. The same criterion was used in both waves.

In the analysis of the former outcome, individuals with presence of depressive episodes at baseline were evaluated in the first follow-up wave, and those presenting depressive episodes at both visits were classified as individuals with maintenance of depressive episodes. The variable was treated dichotomously as maintenance versus non-maintenance of depressive episodes. 

As for the latter outcome, individuals that did not present depressive episodes at baseline were evaluated at the first follow-up wave, and incidence of depressive episodes was defined as those with positive classification for depressive episodes. The variable was treated dichotomously as incidence versus non-incidence of depressive episodes.

#### 2.2.2. Target Exposure

Data on food consumption were obtained with a semiquantitative Food Frequency Questionnaire (FFQ) applied at baseline, consisting of 114 items related to habitual dietary consumption in the previous 12 months. The FFQ is divided into three sections: foods or preparations, amounts in home measurements (portions), and frequency of consumption (“more than 3 times a day”, “2 to 3 times a day”, “once a day”, “5 to 6 times a week”, “2 to 4 times a week”, “once a week”, “1 to 3 times a month”, and “never or almost never”). The amount of each food/preparation in grams (g/d) was calculated as follows: number of portions reported by the participant multiplied by the predefined weight in grams of each portion and the frequency of consumption. Composition of macronutrients and micronutrients of the consumed foods was estimated by the Nutrition Data System for Research software [34].

The validity and reproducibility of the FFQ were assessed previously, and the relative validity was considered reasonable for energy, macronutrients, calcium, potassium, and vitamins E and C, and the reproducibility was satisfactory for all targets [34].

The target dietary components in this article were total polyunsaturated fatty acids (PUFA) and omega-3 fatty acids: alpha-linolenic acid, eicosapentaenoic acid (EPA), docosahexaenoic acid (DHA), and docosapentaenoic acid (DPA). The study used total energy value (TEV) for the descriptive analyses and nutrient adjustment by total calorie intake. 

The references for adequate consumption were those recommended by the *Dietary Reference Intakes* (DRI) of FNB/IOM/DRI (2005) for total energy value, total omega-3, and alpha-linolenic acid and the *International Society for the Study of Fatty Acids and Lipids* (ISSFAL, 2004) for EPA and DHA [35,36].

Data collection on diet was only performed at baseline. In the first follow-up wave, the question *“In the last six months have you changed your eating habits or gone on a diet for any reason?”* allowed proceeding to sensitivity analyses, aimed at assessing the contribution from the reported change in eating habits (not originating from a disease) to the relationship between omega-3 consumption and maintenance and incidence of depression. 

#### 2.2.3. Covariables

The following covariables were used, measured at the follow-up visit: sex (male, female), age categorized by group (39–44 years; 45–64 years; >65 years), schooling (complete primary, complete secondary, complete university), marital status (married/stable union; single; widow(er)/divorced), and race/skin color (white; brown; black; other, i.e., Asian-descendent and indigenous). Monthly per capita family income was categorized in tertiles: 1st tertile, consisting of persons that received up to US$260.43 in 2012; 2nd tertile, including individuals that reported US$260.43 to US$532.73; and 3rd tertile, with individuals with per capita family income greater than US$532.73

Nutritional status was assessed as body mass index (BMI), calculated as $\mathrm{BMI}=\frac{\mathrm{weight}}{{\left(\mathrm{height}\right)}^{2}}$ and classified according to World Health Organization criteria [37] as normal weight (≤24.9 kg/m^2^); overweight (>24.9 kg/m^2^ and ≤29.9 kg/m^2^); and obesity (≥30 kg/m^2^).

In relation to lifestyle variables, we used those corresponding to smoking, grouped as nonsmokers, former smokers, and current smokers; leisure-time physical activity [38], categorized as light (including sedentary individuals), moderate, and vigorous, according to the classification of the International Physical Activity Questionnaire (IPAQ) [39]; and alcohol consumption, categorized as none, light, moderate, or excessive. 

Alcohol consumption was measured as the amount of pure alcohol consumed per week and the frequency of consumption of beer, wine, and distilled liquor. To assess the weekly amount of ethanol in grams, we used the data on consumption of the amounts of pure alcohol in milliliters/week according to the mean alcohol concentration of the above-mentioned beverages, multiplied by the alcohol density (0.8). The resulting value corresponded to the amount of pure ethanol in grams/week, where moderate consumption was defined as less than 210 g/week and 140 g/week, for men and women, respectively, and excessive consumption as 210 g/week and 140 g/week, respectively [40].

#### 2.2.4. Ethical Aspects

Participants in the Longitudinal Study of Adult Health (ELSA-Brasil) signed a free and informed consent form, in compliance with all the ethical principles of privacy and confidentiality of information, and participants’ doubts and questions were answered when necessary. The ELSA-Brasil project was approved by the respective institutional review boards of the six participating research institutions under case reviews 669/06 (USP), 343/06 (FIOCRUZ), 041/06 (UFES), 186/06 (UFMG), 194/06 (UFRGS), and 027/06 (UFBA). The current study was approved by the Institutional Review Board of the Brazilian National School of Public Health (ENSP-FIOCRUZ), under review number 4.831.333.

### 2.3. Statistical Analysis 

Descriptive analyses, means, and standard deviations (SD) were used for the continuous variables and distribution of frequencies for the categorical variables. Wilcoxon’s test was used for the means and χ2 (chi-square) for the categorical variables, with significance set at *p*
≤ 0.05. The nutrients polyunsaturated fatty acids and the omega-3 family were adjusted by total calorie intake using the residual method.

The associations between maintenance and incidence of depression and omega-3 intake were analyzed separately using logistic regression, considering total polyunsaturated fatty acids (PUFA) and the omega-3 family (alpha-linolenic acid, EPA, DHA, DPA).

Variables related to socioeconomic characteristics and health habits that presented *p*-value ≤ 0.01 in the bivariate were included in the multiple regression model. Variables with theoretical clinical and social relevance, such as schooling and nutritional status, even when not reaching *p*
≤ 0.01 in the bivariate analysis (maintenance of depressive episodes), were maintained in the model, considering their epidemiological importance in the relationship between consumption of omega-3 and depression. 

Upon conclusion, we performed a sensitivity analysis, aimed at assessed the contribution from the reported change in eating habits (other than related to treatment of specific chronic diseases) in the relationship between consumption of omega-3 and maintenance and incidence of depression. The final model for each outcome was adjusted by the self-reported changes in eating habits, and possible alterations in the estimates were verified. 

These analyzes were performed due to the possibility of changes in dietary patterns in the last four years of follow-up. Noting that the collection of data related to diet was only performed in the first wave, therefore, it was justified to carry out the aforementioned analysis to verify if the change in the diet pattern could influence the estimates obtained.

The R software version 3.2.2 of 2015 was used for the statistical analyses [41].

## 3. Results

Table 1 shows that variables with higher proportions of maintenance of depressive episodes were: female sex (29.9%), age over 65 years (31.6%), complete primary schooling (32.2%), widow(er)s or divorcees (30.9%), and self-reported black skin color (35.4%). As for lifestyle, smokers (34.0%), nondrinkers (67.5%), and individuals with vigorous physical activity (30.4%) showed the highest proportions of depression maintenance.

Incidence of depressive episodes was higher in women (4.9%), participants 39 to 44 years of age (4.5%), those with complete secondary schooling (4.4%), widow(er)s and divorcees (5.3%), and self-declared Asian-descendant and indigenous individuals (4.6%), with statistical significance (Table 1).

In relation to health behaviors, incidence of depressive episodes was higher among obese individuals, smokers, participants who reported moderate alcohol consumption, and those who practiced light physical activity, with statistical significance (Table 1).

Mean consumption of energy and all the target nutrients was lower in participants with maintenance of depressive episodes, statistically significant for EPA and DHA (Table 2).

In relation to participants who only presented a depressive episode at the first follow-up wave, mean consumption levels of all target nutrients were higher, except for total energy value, with significant differences for omega-3, alpha-linolenic acid, EPA, DHA, and DPA.

Mean energy consumption levels for all the target groups were adequate according to FNB/IOM/DRI guidelines (2400 Kcal) [35]. We also observed that individuals from the four target groups had adequate intake of total omega-3 and alpha-linolenic acid, since the values were greater than the recommended levels (1.6 g for men and 1.1 g for women) [29] (Table 2). In individuals with maintenance depressive episodes and those with incident depression, mean EPA intake was below the level recommended by ISSFAL (2017), while DHA intake was adequate (0.25 g of EPA and 0.25 g of DHA in both sexes) [36].

According to the adjusted model, omega-3 fatty acids have a protective effect against maintenance of depressive episodes, showing important associations, although the significance levels are borderline, possibly due to the sample size. 

As for incidence of depressive episodes, estimates from the adjusted model suggest that the higher the consumption of omega-3 acids (total and subtypes), the lower the risk of developing depressive episodes (protective effect), with significant associations in the consumption of omega-3 and alpha-linolenic acid: OR = 0.91: 95% CI (0.84–0.98) and OR = 0.71: 95% CI (0.59–0.91) (Table 3). 

Sensitivity analysis did not identify significant alterations (greater than 10%) in the estimates after adjusting for changes in eating habits (Table 4).

## 4. Discussion

Our results showed an important protective effect from the consumption of omega-3 (total and subtypes), with a 2% to 65% reduction in the risk of maintenance of depressive episodes, depending on the fatty acid consumed in the omega-3 family, with borderline significant associations. The consumption of total fatty acids and alpha-linolenic acid represented a reduction in the risk of incidence of depressive episodes (9% and 29%, respectively). The main sources of this nutrient are fish in general, especially sardines, salmon, and tuna [42]

Maintenance of depressive episodes in our study was 27.1%, but due to the scarcity of studies, we did not find any Brazilian estimates for comparison. However, Horiwaka et al., 2018, in a study on the association between omega-3 intake and depressive symptoms in a Japanese cohort, found maintenance of depressive symptoms in 23.04% of the eligible participants, less than in our population [43]. This difference may be due to differences in the instruments for assessing depression, as well as eating habits, such as the greater consumption of omega-3 food sources, especially fish, by the Japanese population.

Incidence of depression among participants in ELSA-Brasil (3.7%) was lower than in two Canadian cohorts in 2017 (6.6%) and 2014 (12.1%) [44,45], a Swedish cohort in 2016 (12.3%) [46], and in the world population in 2015 (12.4%) [3]. The differences found by our study can be attributed to the fact that our study population was working, and was thus less prone to depression, and to the use of different instruments for the assessment of depressive episodes (our study used CIS-R, which assesses these events in the previous seven days).

Some cross-sectional, ecological, and prospective studies have analyzed the association between depression and consumption of omega-3 fatty acids, showing this nutrient’s role in adequate functioning of the central nervous system, in the inflammatory chain, and in disease mechanisms [15,16,40]. According to McNamara [22] and Wani et al. [6] translational epidemiological evidence suggests that deficient dietary omega-3 intake is a modifiable risk factor for depression, and that individuals with low consumption of omega-3 food sources display increased depressive symptoms. However, the results of this relationship are inconsistent, since other studies have failed to find a significant association between consumption of these polyunsaturated fatty acids and their benefits in the prevention and treatment of depression [15,27,47]. Among the hypotheses for this inconsistency, some authors point to differences between study designs, clinical issues, methods for assessment and diagnosis of depression, environmental variability, possible selection bias, and methodological differences in the studies [15,27,46,47].

Our study found that consumption of all types of omega-3 fatty acids was lower among individuals with maintenance of depressive episodes. As for individuals with incident depression, the mean intake of all types of omega-3 was consistent with that of individuals without depressive episodes in either wave. Araújo et al., 2020 also observed lower consumption of omega-3 food sources in depressed individuals treated at a psychology clinic [48]. Energy intake was also higher in participants who only presented depressive episode in the second wave.

Despite the majority of participants having adequate consumption of omega-3, the dietary intake of this nutrient by the western population has drastically reduced during the last century, with a concomitant worsening of the food quality of the world population accompanying an increase of mental illnesses [43,49,50,51]. Some authors credit the increased prevalence of depression to the stress of modern life, the increased intake of pro-inflammatory foods and nutrients such as saturated fats, and the reduced intake of ancestral foods such as fruits, vegetables, fish, and seafood [43,50,51]. The Mediterranean diet, being balanced and rich in fresh fruits, vegetables, bioactive compound oils, whole grains, and fish, has been shown to be an important therapeutic resource in the fight against several diseases, including depression [43,49,51].

Horikawa et al., 2018, in their study of the NILS-LSA cohort, found that higher intake of EPA and DHA was effective in reducing depressive symptoms in Japanese individuals 40–74 years of age [43], consistent with our findings of an inverse relationship between consumption of alpha-linolenic acid, EPA, PUFA, omega-3, DHA, and DPA and the maintenance of depressive episodes. Zhang et al., 2020, using data from the *National Health and Nutritional Examination Surveys* (NHANES), found a similar relationship with omega-3 intake as in our study, where consumption was inversely associated with risk of depressive symptoms [52]. Zhang et al., 2020 and Horikawa et al., 2018 found associations that mirror our findings in relation to maintenance of depressive episodes after adjustments [43,52].

As for incidence of depression, Lai et al. (2016) [53], Horikawa et al. [43], Yang et al. [47], and Zhang et al. [52] reported similar results to those of our study, with the consumption of omega-3, EPA, DHA, DPA, alpha-linolenic acid, and PUFA inversely associated with the appearance of depressive episodes.

The current study’s strengths feature its originality, as it is the first to assess associations between maintenance and incidence of depressive episodes and consumption of omega-3, besides the use of data from the ELSA-Brasil Study, with rigorous data collection protocols and reliable and validated instruments, thus guaranteeing the quality of the sample and the data.

The first limitation to the study is the fact that the ELSA-Brasil sample consists only of public employees, with the potential for a selection bias (healthy worker phenomenon). Although the food frequency questionnaire is a low-cost and simple method, its potential limitations are the underestimation of daily intake of foods and dependence on recall, possibly leading to a differential classification bias. However, no instrument exists for assessing food intake without some limitations. However, the results from the FFQ were validated and tested for reliability in ELSA-Brasil by the intra-class correlation coefficient (ICC) for total lipids, however, it’s reliablility as specifically regards omega-3 is questionable. The second limitation is the absence of the application of the food frequency questionnaire in the follow-up wave (2012–2014). According to the sensitivity analysis, the changes in dietary habits (unrelated to certain diseases) reported by study participants appeared not to have been large enough to alter the findings. The estimates did not change more than 10%, and the largest alterations were observed between the consumption of EPA and the maintenance of depressive episodes, and between the consumption of DHA and the odds of incident depression, where the OR changed by 1% in relation to the previous value after the adjustment. Importantly, the FFQ can estimate habitual intake, and few dietary changes were observed in the four years between baseline and the follow-up visit, due to a possible small variation in eating patterns among the participants. 

The third limitation would be the use of BMI as an indicator of nutritional status, while not being indicative of omega-3 status. The serum omega-3 level would provide a better indicator of omega-3 status. However, the objective of the study is to assess the intake, not the serum level of adequacy, by which several pathological and physiological issues not addressed can be influenced.

The last limitation is inherent to the CIS-R; that is, despite permitting classification of depressive episodes according to the ICD-10, allowing an approximation of the diagnosis, the CIS-R only assesses these symptoms over the previous seven days, without considering prior assessment of depression or its symptoms. This peculiarity hinders the understanding of chronic depression and the differentiation of individuals with depressive episodes, that is, whether they are only experiencing the symptoms at the time of the data collections or are having recurrent depressive episodes. In addition, the instrument may not capture asymptomatic chronically depressed participants due to possible waning of symptoms during the week of data collection for ELSA-Brasil. 

The study’s results suggest that dietary consumption of omega-3, DHA, EPA, DPA, and alpha-linolenic acid may have a protective effect against the maintenance and incidence of depressive episodes.

## Figures and Tables

**Figure 1 nutrients-14-03227-f001:**
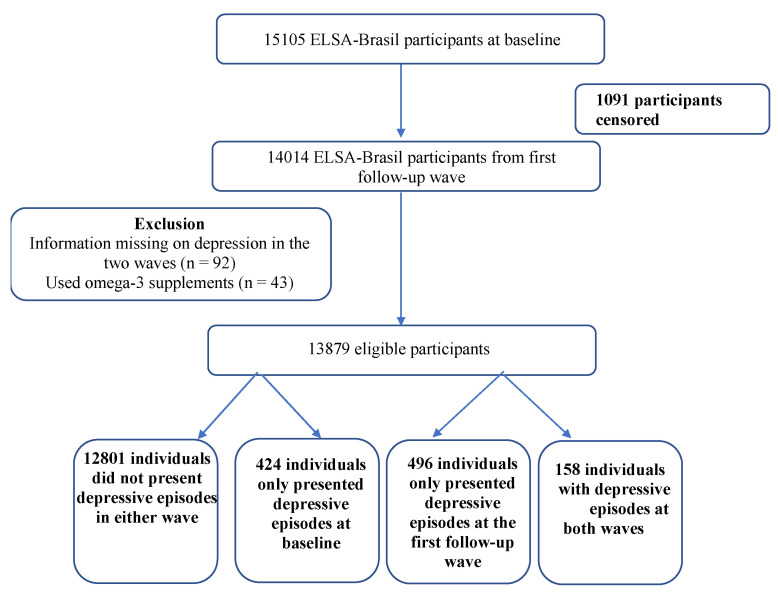
Final study population (*n* = 13.879, ELSA-Brasil, 2008–2014).

**Table 1 nutrients-14-03227-t001:** Socioeconomic characteristics and health behaviors of participants in the ELSA-Brasil Study according to maintenance and incidence of depressive episodes, 2008–2010 (baseline) and 2012–2014 (first follow-up wave).

Variables	Maintenance Depressive Episodes *n* (%)	Non- Maintenance of Depressive Episodes *n* (%)	*p*-Value ^a^	Incidence of Depressive Episodes *n* (%)	Non- Incidence of Depressive Episodes *n* (%)	*p*-Value ^a^
**Total**	158 (27.1)	424 (72.9)	<0.001 *	496 (3.7)	12,801 (96.3)	<0.001 *
**Sex**						
Male	27 (18.6)	118 (81.4)	0.004 *	147 (2.4)	6021 (97.6)	<0.001 *
Female	131 (29.9)	306 (70.1)		349 (4.9)	6780(95.1)	
**Age group**						
39–44	13 (24.5)	40 (75.5)	0.65	62 (4.5)	1309 (95.5)	<0.001 *
45–64	121 (26.7)	332 (73.3)		378 (4.0)	9179 (96.0)	
>65 years	24 (31.6)	52 (68.4)		56 (2.4)	2313 (97.6)	
**Schooling**						
Complete primary	28 (32.2)	59 (67.8)	0.03 *	64 (4.3)	1433 (95.7)	0.01 *
Complete secondary	76 (30.8)	171 (69.2)		183 (4.4)	3939 (95.6)	
Complete university	54 (21.8)	194 (78.2)		249 (3.2)	7429 (96.8)	
**Marital status**						
Married/stable union	77 (25.4)	226 (74.6)	0.60	266 (3.1)	8364 (96.9)	<0.001 *
Single	26 (25.7)	75 (74.3)		82 (4.3)	1818 (95.7)	
Widow(er)/Divorced	55 (30.9)	123 (69.1)		148 (5.3)	2619 (94.7)	
**Race/Color**						
White	62 (23.7)	200 (76.3)	0.04 *	225 (3.2)	6710 (96.8)	0.02 *
Brown	50 (26.9)	136 (73.1)		148 (4.1)	3499 (95.9)	
Black	40 (35.4)	71 (64.6)		94 (4.5)	1997 (95.5)	
Other ^b^	6 (26.1)	17 (73.9)		29 (4.6)	595 (95.4)	
**Monthly per capita family income**						
1st tertile	45 (25.6)	131 (74.4)	0,6	150 (3.6)	3995 (96.4)	0.06
2nd tertile	60 (29.9)	141 (70.1)		158 (3.6)	4191 (96.4)	
3rd tertile	53 (25.9)	152 (74.1)		170 (3.8)	4283 (96.2)	
Missing	-	-		18 (5.1)	332 (94.9)	
**Nutritional status**						
Normal weight	52 (34.0)	101 (66.0)	0.14	140 (3.2)	4205 (96.8)	<0.001 *
Overweight	49 (22.8)	166 (77.2)		184 (3.3)	5317 (96.7)	
Obesity	57 (26.6)	157 (73.4)		172 (5.0)	3279 (95.0)	
**Smoking**						
Smoker	31 (31.3)	68 (68.7)	0.59	79 (5.6)	1328 (94.4)	0.01 *
Former smoker	43 (25.0)	129 (75.0)		141 (3.5)	3932 (96.5)	
Nonsmoker	84 (27.0)	227 (73.0)		276 (3.5)	7541 (96.5)	
**Alcohol consumption**						
None	85 (67.5)	41 (32.5)	0.27	63 (3.7)	1643 (96.3)	0.02 *
Light	55 (24.3)	171 (75.7)		204 (3.4)	5838 (96.6)	
Moderate	11 (5.4)	194(94.6)		201 (4.4)	4372 (95.6)	
Excessive	7 (28.0)	18 (72.0)		28 (2.9)	948 (97.1)	
**Physical activity**						
Light	143 (29.1)	348 (70.9)	0.08	420 (4.3)	9391 (95.7)	<0.001 *
Moderate	8 (11.8)	60 (88.2)		56 (2.3)	2334 (97.7)	
Vigorous	7 (30.4)	16 (69.6)		20 (1.8)	1076 (98.2)	

Note: (a) *x^2^* test for differences in proportions; (b) other: Asian-descendant and indigenous; (*) significance at *p*-value ≤ 0.05.

**Table 2 nutrients-14-03227-t002:** Mean daily dietary intake by participants with/without maintenance of depressive episodes and with/without incident depression in the ELSA-Brasil Study, 2008–2010 (baseline) and 2012–2014 (first follow-up wave).

Indicator	Maintenance Depressive Episodes *n* = 158	Non-Maintenance of Depressive Episodes *n* = 424	*p*-value ^b^	Incidence of Depressive Episodes *n* = 496	Non-Incidence of Depressive Episodes *n* = 12,801	*p*-Value ^b^
	Mean (Standard Deviation)			Mean (Standard Deviation)		
Total energy value (kcal)	2665.2 (1171.8)	2800.1 (1109.0)	0.09	2834.7 (1266.5)	2658.5 (1049.2)	0.004
Omega-3 (g) ^a^	2.7 (1.1)	2.9 (1.2)	0.09	2.7 (1.2)	2.9 (1.2)	0.001 *
Alpha-linolenic acid (g) ^a^	1.9 (0.5)	2.0 (0.5)	0.6	1.9 (0.5)	2.0 (0.4)	0.004 *
Polyunsaturated fat (PUFA) (g) ^a^	16.2 (4.1)	16.5 (3.7)	0.38	16.3 (3.9)	16.6 (3.6)	0.06
Eicosapentaenoic acid (EPA) (g) ^a^	0.13 (0.2)	0.16 (0.2)	<0.001 *	0.16 (0.2)	0.17 (0.2)	0.002 *
Docosahexaenoic acid (DHA) (g) ^a^	0.44 (0.5)	0.54 (0.6)	0.05 *	0.52 (0.6)	0.60 (0.6)	0.002 *
Docosapentaenoic acid (DPA) (g) ^a^	0.12 (0.1)	0.15 (0.2)	0.06	0.13 (0.2)	0.15 (0.2)	0.003 *

Note: (^a^) target exposures; (^b^) Wilcoxon’s test; (*) significance at *p*-value ≤ 0.05.

**Table 3 nutrients-14-03227-t003:** Odds ratios (OR) and confidence intervals for associations between maintenance of depressive episodes, incidence of depression, and consumption of omega-3, alpha-linolenic acid, polyunsaturated fats (PUFA), eicosapentaenoic acid (EPA), docosahexaenoic acid (DHA), and docosapentaenoic acid (DPA) with the estimates from models 2 and 3. ELSA-Brasil Study 2008–2010 (baseline) and 2012–2014 (first follow-up visit).

Maintenance of Depressive Episodes	Crude OR (95% CI)	Model 1 OR (95% CI)	Model 2 OR (95% CI)
**Omega-3**	0.88 (0.73–1.03)	0.87 (0.73–1.02)	0.87 (0.70–1.03) **
**Alpha-linolenic acid**	0.91 (0.61–1.35)	0.91 (0.61–1.37)	0.93 (0.62–1.39)
**EPA**	0.36 (0.10–1.11)	0.35 (0.10–1.07)	0.35 (0.10–1.09)
**DHA**	0.73 (0.50–1.02)	0.71 (0.49–1.05)	0.72 (0.49–1.01) **
**DPA**	0.32 (0.10–1.17)	0.29 (0.07–1.10)	0.31 (0.10–1.13)
**PUFA**	0.98 (0.93–1.02)	0.98 (0.94–1.03)	0.98 (0.94–1.04) **
**Incidence of Depressive Episodes**	**Crude** **OR (95% CI)**	**Model 1 ^1^** **OR (95% CI)**	**Model 2 ^1^** **OR (95% CI)**
**Omega-3**	0.90 (0.82–0.98 )*	0.91 (0.82–0.98) *	0.91 (0.84–0.98) *
**Alpha-linolenic acid**	0.72 (0.59–0.89) *	0.73 (0.59–0.90) *	0.71 (0.59–0.91) *
**EPA**	0.62 (0.34–1.09)	0.67 (0.37–1.15)	0.69 (0.39–1.20)
**DHA**	0.87 (0.73 -1.02) **	0.88 (0.74 -1.04) **	0.89 (0.75 -1.05) **
**DPA**	0.59 (0.29–1.12)	0.62 (0.32–1.18)	0.66 (0.33–1.25)
**PUFA**	0.97 (0.95–1.00) **	0.98 (0.95–1.00) **	0.98 (0.95–1.00) **

Model 1: sex + schooling; Model 2: model 1 + nutritional status. Model 1^1^: sex + schooling + marital status + age group; Model 2^1^: model 1^1^ + nutritional status + smoking + physical activity. (*) significant (**) relevant magnitudes, but with borderline significance.

**Table 4 nutrients-14-03227-t004:** Sensitivity analyses adjusted for changes in eating habits. ELSA-Brasil 2012 Study, 2014 (first follow-up visit).

	Maintenance of Depressive Episodes ^a^ OR (95% CI)	Incidence of Depressive Episodes ^b^ OR (95% CI)
**Omega-3**	0.87 (0.73–1.03)	0.91 (0.83–0.99)
**alpha-linolenic acid**	0.93 (0.62–1.38)	0.74 (0.60–0.91)
**EPA**	0.35 (0.10–1.08)	0.69 (0.38–1.20)
**DHA**	0.72 (0.49–1.00)	0.89 (0.75–1.05)
**DPA**	0.30 (0.07–1.11)	0.65 (0.33–1.23)
**PUFA**	0.99 (0.94–1.04)	0.97 (0.95–0.99)

Notes: (a): Model 1: Sex + schooling + total energy value + nutritional status + changes in eating habits; (b) Model 2: sex + schooling + nutritional status + smoking + physical activity + marital status + age bracket.

## Data Availability

The data presented in this study are available on request from the corresponding author. The data are not publicly available due to commitment to maintain confidentiality and secrecy of the database made available by the ELSA-Brasil Project, as well as all information related to the project entitled.

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
