# Peer review of "Consumption of Omega-3 and Maintenance and Incidence of Depressive Episodes: The ELSA-Brasil Study"

_nutrients, 2022, doi:10.3390/nu14153227_

Round 1

Reviewer 1 Report

Dear Authors,

It would be useful/necessary to have much more precision on the calculations and expression of ranges of omega-3 quantity from food questionnaires.

Also to present at least partially the questions and presentations of questionnaires of the studied that lead to this analysis.

Also add a rationale with references that explain a bit more the biochemical link between depression and oemga-3 (ie anti-inflammatory/antioxidative process, others).

Author Response

COMMENTS FROM THE REVIEW 1:

MATERIAL AND METHODS

REVIEW 1

  1. Also to present at least partially the questions and presentations of questionnaires of the studied that lead to this analysis.
  2. It would be useful/necessary to have much more precision on the calculations and expression of ranges of omega-3 quantity from food questionnaires.

ANSWER: Dear review, in methods we describe the questions and presentations of FFQ and CIS-R of the studied that lead to this analysis, more precision on the calculations and expression of ranges of omega-3 quantity and in results we describe the adequate ranges of omega 3 by ISSFAL.

FFQ

Data on food consumption were obtained with a semiquantitative Food Frequency Questionnaire (FFQ) applied at baseline and consisting of 114 items related to habitual dietary consumption in the previous 12 months. The FFQ is divided into three sections: foods or preparations, amounts in home measurements (portions), and frequency of consumption (“more than 3 times a day”, “2 to 3 times a day”, “once a day”, “5 to 6 times a week”, “2 to 4 times a week”, “once a week”, “1 to 3 times a month”, and “never or almost never”). The amount of each food/preparation in grams (g/d) was calculated as follows: number of portions reported by the participant multiplied by the predefined weight in grams of each portion and the frequency of consumption. Composition of macronutrients and micronutrients of the consumed foods was estimated by the Nutrition Data System for Research software [35].

RECOMENDATION

Mean energy consumption levels for all the target groups were adequate according to FNB/IOM/DRI guidelines (2,400 Kcal) [36]. We also observed that individuals from the four target groups had adequate intake of total omega-3 and alpha-linolenic acid, since the values were greater than the recommended levels (1.6 grams for men and 1.1 grams for women) [30] (Table 2). In individuals with persistent depressive episodes and those with incident depression, mean EPA intake was below the level recommended by ISSFAL (2017), while DHA intake was adequate (0.25 grams of EPA and 0.25 grams of DHA in both sexes) [37].

CIS-R

The instrument was translated and adapted to Brazilian Portuguese and consists of 14 sections corresponding to symptoms that caused suffering and alterations in routine activities in the previous week: anxiety, phobia, panic, compulsions, obsessions, physical symptoms, fatigue, depression, depressive ideas, irritability, lack of concentration and forgetfulness, altered sleep, preoccupation with body functioning, and general preoccupations [34].

The CIS-R was applied by trained and certified interviewers, in face to face interview as part of the whole ELSA-Brasil questionnaire. Depressive episodes were computed by an algorithm developed by Lewis et al. according to ICD-10 criteria (F32.xx) for depressive episodes (mild, moderate with symptoms, moderate without symptoms, severe with symptoms, severe without symptoms) [34]. These depressive episodes classified by the CIS-R were grouped in a dichotomous variable as presence or absence of depressive episode. There were no clinical diagnoses for depression assessed by clinicians.

INTRODUCTION AND DISCUSSION

REVIEW 1

Also add a rationale with references that explain a bit more the biochemical link between depression and omega-3 (ie anti-inflammatory/antioxidative process, others).

ANSWER: In the introduction, we have included three paragraphs on the biochemistry and mechanisms of action of fats.

Omega 3 fatty acids have several functions in the body and metabolism, being a disease protection factor due to their anti-inflammatory, antithrombotic and anti-arteriosclerotic effects, also acting on the modulation of serotonin and allowing the increase of the availability of this neurotransmitter in the synaptic cleft, essential for neural functioning [17,18-19]. Changes in the action, availability and serum production of serotonin and dopamine are related to the pathophysiology of depression, although all the mechanisms involved are not elucidated [14,20].

Some of these mechanisms listed in the literature explain the relationship between omega 3 and depression. Among them, it is highlighted that despite the reduction in the amount of these neurotransmitters mentioned above due to the reduction of neuromodulation arising from the low serum level of omega 3 fatty acids, the actions of the reuptake pump and the enzymes involved in the degradation of these substances remain unchanged. Consequently, in depression there is a lower uptake of neurotransmitters by the receptor neuron and a functioning of the Central Nervous System (CNS) with an inadequate level of these [21].

In addition, this imbalance may come from a deregulation of neurotransmission due to the instability of the neural cell membrane due to omega 3 deficit [22]. Generally, this destabilization occurs in situations of stress or inflammation, where some enzymes are attached to the cell membrane and remove fatty acids from phospholipids, deregulating serotonin uptake receptors and other neurotransmitters [22,23]..

In the discussion section, we included dietary sources of omega 3.

The main sources of this nutrient are fish in general, especially sardines, salmon and tuna [43].

Reviewer 2 Report

Thank you for the opportunity to review this paper, the study is interesting and adds to the knowledge about omega 3 and mood disorders, however there are issues to be addressed that would improve the paper and I suggest major revision.

Abstract - young and old adults these are imprecise terms and open to varying interpretations so it is best to specify age ranges - data was I think gathered on 30-44s and 45-64s and >65s . Young adults can be interpreted as 18-24, so I recommend clarification of this in the abstract to avoid confusion.

Introduction - The introduction is very brief, only 24 lines  (lines 42-66) and hence seems to only skim the surface. It needs to be rewritten and expanded. The omega 3 fats should be introduced individually in more detail and food sources outlined.  Metabolism of omega 3 fats is reliant on cofactors (including minerals) and negatively affected in diabetes and atopic conditions and this should be mentioned given the prevalence of those conditions and as they are frequently co-morbid with depression. Mechanisms of action of omega 3 fats are only mentioned very briefly and I think more detail should be given regarding each. I also think there should be brief mention of data meta analysis and RCTs on omega 3 supplementation and depression - including levels of intake (dosage) as this provides useful contextual data for considering mean intakes of omega 3 reported in results.

Materials and methods - please clarify if medically diagnosed depression or self reported depression?

It is good that the FFQ was validated, however it is not clear it it  was it assessed as being reliable for omega 3 specifically (line 153)? Please clarify.

Line 167 - methods used for sensitivity analysis needs more detail - please provide details to make reproducible, how did you calculate if this impacted intake of omega 3?

180 - BMI as indicator of nutritional status - this is only a very superficial indicator - and not in any way indicative of omega 3 status (red cell omega 3 EFA status) would have provided a much clearer picture and enhanced methods - this should be considered in limitations and discussion.

Results - generally well presented. Reports sufficient intake in all groups in line 248 - in the discussion this needs further evaluation and consideration of optimal amounts and perhaps to also consider wider context such as amounts in ancestral human diet.

Table 1 variables - Use of terms Brown Black is questionable.  Should this be Asian/Hispanic origin or Afro-Caribbean?

Author Response

COMMENTS FROM THE REVIEW 2:

REVIEW 2

ABSTRACT

young and old adults these are imprecise terms and open to varying interpretations so it is best to specify age ranges - data was I think gathered on 30-44s and 45-64s and >65s .Young adults can be interpreted as 18-24, so I recommend clarification of this in the abstract to avoid confusion.

ANSWER: We changed the terms as suggested.

The study aims to assess the association between the consumption of omega-3 and the maintenance and incidence of depressive episodes in adults (39-64 years) and elderly adults (>65 years).

REVIEW 2

INTRODUCTION:

The introduction is very brief, only 24 lines (lines 42-66) and hence seems to only skim the surface. It needs to be rewritten and expanded. The omega 3 fats should be introduced individually in more detail and food sources outlined.  Metabolism of omega 3 fats is reliant on cofactors (including minerals) and negatively affected in diabetes and atopic conditions and this should be mentioned given the prevalence of those conditions and as they are frequently co-morbid with depression. Mechanisms of action of omega 3 fats are only mentioned very briefly and I think more detail should be given regarding each. I also think there should be brief mention of data meta-analysis and RCTs on omega 3 supplementation and depression - including levels of intake (dosage) as this provides useful contextual data for considering mean intakes of omega 3 reported in results.

ANSWER: In the introduction, we have included three paragraphs on the biochemistry and mechanisms of action of fats.

Omega 3 fatty acids have several functions in the body and metabolism, being a disease protection factor due to their anti-inflammatory, antithrombotic and anti-arteriosclerotic effects, also acting on the modulation of serotonin and allowing the increase of the availability of this neurotransmitter in the synaptic cleft, essential for neural functioning [17,18-19]. Changes in the action, availability and serum production of serotonin and dopamine are related to the pathophysiology of depression, although all the mechanisms involved are not elucidated [14,20].

Some of these mechanisms listed in the literature explain the relationship between omega 3 and depression. Among them, it is highlighted that despite the reduction in the amount of these neurotransmitters mentioned above due to the reduction of neuromodulation arising from the low serum level of omega 3 fatty acids, the actions of the reuptake pump and the enzymes involved in the degradation of these substances remain unchanged. Consequently, in depression there is a lower uptake of neurotransmitters by the receptor neuron and a functioning of the Central Nervous System (CNS) with an inadequate level of these [21].

In addition, this imbalance may come from a deregulation of neurotransmission due to the instability of the neural cell membrane due to omega 3 deficit [22]. Generally, this destabilization occurs in situations of stress or inflammation, where some enzymes are attached to the cell membrane and remove fatty acids from phospholipids, deregulating serotonin uptake receptors and other neurotransmitters [22,23].

In discussion, we included dietary sources of omega 3.

The main sources of this nutrient are fish in general, especially sardines, salmon and tuna [43].

Although we understand the relevance of the suggestion regarding the cofactors and morbidities that influence omega 3 metabolism, these variables were not explored in our analyses.

REVIEW 2

MATERIAL AND METHODS

Please clarify if medically diagnosed depression or self reported depression?

ANSWER: Thank you for your suggestion. We inserted the information about depression diagnoses expanding the CIS-R description as below:

The instrument was translated and adapted to Brazilian Portuguese and consists of 14 sections corresponding to symptoms that caused suffering and alterations in routine activities in the previous week: anxiety, phobia, panic, compulsions, obsessions, physical symptoms, fatigue, depression, depressive ideas, irritability, lack of concentration and forgetfulness, altered sleep, preoccupation with body functioning, and general preoccupations [34].

The CIS-R was applied by trained and certified interviewers, in face to face interview as part of the whole ELSA-Brasil questionnaire. Depressive episodes were computed by an algorithm developed by Lewis et al. according to ICD-10 criteria (F32.xx) for depressive episodes (mild, moderate with symptoms, moderate without symptoms, severe with symptoms, severe without symptoms) [34]. These depressive episodes classified by the CIS-R were grouped in a dichotomous variable as presence or absence of depressive episode. There were no clinical diagnoses for depression assessed by clinicians.

It is good that the FFQ was validated, however it is not clear it it  was it assessed as being reliable for omega 3 specifically (line 153)? Please clarify.

ANSWER: The reproducibility and validity were assessed by the intra-class correlation coefficient (ICC) for lipids total,but it´s not to reliable for omega 3 specifically.

Line 167 - methods used for sensitivity analysis needs more detail - please provide details to make reproducible, how did you calculate if this impacted intake of omega 3?

ANSWER: In methods, we included the text below.

These analyzes were performed due to the possibility of changes in dietary patterns in the last four years of follow-up. Noting that the collection of data related to the diet was only performed in the first wave, therefore, it is justified to carry out the aforementioned analysis to verify if the change in the diet pattern can influence the estimates obtained.

180 - BMI as indicator of nutritional status - this is only a very superficial indicator - and not in any way indicative of omega 3 status (red cell omega 3 EFA status) would have provided a much clearer picture and enhanced methods - this should be considered in limitations and discussion.

ANSWER: In discussion, we included the text below.

Third limitation would be the use of BMI as an indicator of nutritional status, but not indicative of omega 3 status. The serum omega 3 level would provide a better indicator of omega 3 status. However, the objective of the study is to assess intake and not of the serum level of adequacy, which several pathological and physiological issues not addressed can be influenced.

REVIEW 2

RESULTS

Generally well presented. Reports sufficient intake in all groups in line 248 - in the discussion this needs further evaluation and consideration of optimal amounts and perhaps to also consider wider context such as amounts in ancestral human diet.

ANSWER: Thank you for your suggestion. In discussion, we included the text below.

Despite the majority of participants having adequate consumption of omega 3, the dietary intake of this nutrient by the western population has drastically reduced during the last century, with a concomitant worsening of the food quality of the world population with the increase of mental illnesses [44,50-52]. Some authors credit the increased prevalence of depression to the stress of modern life, the increased intake of pro-inflammatory foods and nutrients such as saturated fats, and the reduced intake of ancestral foods such as fruits, vegetables, fish and seafood [44,51-52]. The Mediterranean diet, being balanced and rich in fresh fruits, vegetables, bioactive compound oils, whole grains and fish, has been shown to be an important therapeutic resource in the fight against several diseases, including depression [44,50;52].

Table 1 variables - Use of terms Brown Black is questionable.  Should this be Asian/Hispanic origin or Afro-Caribbean?

ANSWER: The race/skin color classification of the ELSA-Brasil articles follows the IBGE (the Brazilian Institute of Statistics and Geography, responsible for the Brazilian Census) categories (white; brown (“pardo”); black; Asian-descendent; indigenous). Additionally, it is a self-declared race/skin color collected during de ELSA-Brasil face to face interviews.